# Public Attitude towards Biobanking: An Italian University Survey

**DOI:** 10.3390/ijerph192013041

**Published:** 2022-10-11

**Authors:** Chiara Aleni, Carmela Rinaldi, Valentina Bettio, Eleonora Mazzucco, Annamaria Antona, Cristina Meini, Emiliano Loria, Paolo Bonvicini, Silvia Vittoria Cracas, Silvia Caristia, Antonio Rimedio, Fabrizio Faggiano, Daniela Ferrante, Daniela Capello

**Affiliations:** 1Department of Sustainable Development and Ecological Transition, University of Piemonte Orientale, 13100 Vercelli, Italy; 2“Maggiore della Carità” Hospital, 28100 Novara, Italy; 3UPO Biobank, University of Piemonte Orientale, 28100 Novara, Italy; 4Department of Translational Medicine, University of Piemonte Orientale, 28100 Novara, Italy; 5Biobanks and Complex Data Management, University of Cote d’Azur, 06108 Nice, France; 6Ethics Committee of the “Maggiore della Carità” Hospital, 28100 Novara, Italy

**Keywords:** biobanks, survey, population

## Abstract

Biobanks have established a critical role in biomedical research by collecting, preserving, organizing, and disseminating biospecimens and related health data, contributing to precision medicine development. Participation in biobanks is influenced by several factors, such as trust in institutions and scientists, knowledge about biobanking, and the consideration of benefit sharing. Understanding public attitudes, fears, and concerns toward biobanking is fundamental to designing targeted interventions to increase trust towards biobanks. The aim of our study was to investigate the level of knowledge and perception of biobanks in students and personnel of the University of Piemonte Orientale. An online questionnaire was designed and administered via e-mail. A total of 17,758 UPO personnel and students were invited to participate in the survey, and 1521 (9.3%) subjects completed the survey. The results showed that 65.0% of the participants were aware of the term “biobank” and knew what the activity of a biobank was, and 76.3% of subjects were willing to provide biospecimens to a research biobank, whereas 67.3% of the respondents were willing to contribute, in addition to biospecimens, their health and lifestyle data. Concerns were raised about the confidentiality of the information (25.6%) and the commercial use of the samples (25.0%). In conclusion, participants were aware of the role that biobanks play in research and were eager to participate for the sake of furthering scientific research. Still, several concerns need to be addressed regarding the confidentiality of the data along with the commercial use of the samples and associated data.

## 1. Introduction

Personalized and precision medicine have acquired increasing attention over recent years. Advances in molecular medicine, genetics, and bioinformatics have made it possible to optimize prevention strategies, identify and tailor therapies, and determine predisposition to disease [1,2]. The success of personalized medicine requires a critical number of biological, genetic, health, and other personal data (i.e., lifestyle and socioeconomic data) organized according to scientific criteria and high-quality standards [3,4]. Research biobanks respond to this need by collecting, storing, and disseminating biospecimens along with the associated information to sustain high-impact biomedical research [5].

Human research biobanks can be distinguished according to the types of biological samples, such as DNA, plasma, and tissue samples, or the purpose, including disease-based (samples from patients with a specific disease), population-based, or project-driven biobanks [6]. Population biobanks are a fundamental tool to carry out pioneering multidisciplinary research for the investigation of genetic, environmental, lifestyle, and socioeconomic factors associated with the state of health [7]. The functioning and the success of any biobank require the participation of many individuals willing to provide data and/or biological material [3]. Participation is influenced by many factors, such as the participant’s knowledge about biobanking [8,9], trust in research institutions and scientists [9], consideration of benefit sharing [10], expected benefits, and personal beliefs (cultural and religious) [11]. At the same time, biobank activity entails several challenges, including the recruitment of the participants, data security, informed consent, data sharing, future research topics, the returning of results, and other legal, social, and ethical issues [12]. Understanding the public’s attitude toward biobanking and the engagement of the population are key factors to developing more effective recruitment strategies [3,13]. Overall, studies published to date demonstrate that knowledge and perception of biobanks are still limited [3], and consequently understanding fears and concerns about the reliance on personal data and biological samples is fundamental for designing targeted interventions to spread information and awareness and increase trust toward biobanks.

The UPO Biobank is the institutional research biobank of the University of Eastern Piedmont (UPO). It was implemented in April 2020 and is dedicated to supporting aging research. The UPO Biobank was established as a multispecialty biobank with both a population- and disease-oriented commitment.

The aims of this study were to investigate the level of knowledge and perception of biobanks in students and personnel of the University of Eastern Piedmont, focusing on the perceived benefits and risks of biobanking and the willingness to participate and to test and optimize a survey aimed at investigating the perception and willingness to donate in the general population.

## 2. Materials and Methods

### 2.1. Study Design and Population

This was a cross-sectional, survey-based study. A questionnaire-based survey was conducted among students and personnel of the University of Eastern Piedmont to explore attitudes, concerns, and expectations towards biobanking and the willingness to provide samples/data for biobank activity. Around 17,000 students and university personnel were invited to participate in the survey via e-mail between August 2021 and December 2021. Participants were over 18, including students enrolled in any course and personnel (including administration and academics) of the University of Eastern Piedmont, and they had to sign the informed consent. The exclusion criterion was the absence of signed informed consent. The study protocol was approved by the UPO data protection officer and the local ethical committee.

### 2.2. Study Tool and Data Collection

An online Italian-language survey was implemented using the REDCap platform (REDCap, Vanderbilt University). The questionnaire consisted of three main sections: (a) sociodemographic data and information about trust in research and researchers; (b) knowledge about biobanks and their activities; and (c) a UPO Biobank activity description and willingness to participate in it. Questionnaire validation and improvement was carried out before the survey launch according to 10 researchers’ feedback and evaluations.

Participants were invited to participate in the survey via e-mail. Those who decided to participate were invited to click a link in the e-mail to be redirected to the REDCap platform, where they found a short introductory text explaining how the data would be collected, stored, and anonymized. The participant’s informed consent for data processing was mandatory to access the questionnaire section.

### 2.3. Measures and Questionnaire

*Demographic Data.* In this section, containing 22 items, participants were asked about their age, sex, nationality, role in the University (student, PhD student, research fellow, professor, or administrative/technical staff), their department of affiliation, and information about the family (occupation of father/mother). Moreover, information about the health status of the participants was collected (general health status and diagnosis of chronic conditions). Participants were also asked to rate their interest in scientific subjects, their knowledge of biomedical subjects, and their trust in scientists and research. As the survey was conducted during the peak of the second wave of the COVID-19 pandemic, participants were also asked about their interest and trust in science before and after the pandemic.

*Biobanks and their activities*. In this section, containing 10 items, participants were asked about their knowledge of biobank activity. Subjects were also interviewed about their willingness to provide biospecimens and personal data to biobanks and about concerns, including a lack of confidence in the usefulness of their sample or in the quality of the research, a fear that the sample would be used for commercial purposes, a fear of genetic research, a fear of needles, and the time spent participating. Furthermore, participants were questioned about their willingness to provide their personal data and what type of consent they were willing to provide.

*Knowledge about the UPO Biobank*. In the last section of the questionnaire, containing two items, participants were asked about their willingness to participate in UPO Biobank projects by providing biospecimens and personal data.

### 2.4. Statistical Analysis

Quantitative variables are presented as the median and interquartile range (IQR) because the data were not normally distributed. Categorical variables are summarized as counts and percentages. Differences in medians were evaluated using the Mann–Whitney test. Associations between categorical variables were tested using the Pearson χ^2^ test. A two-sided *p* value < 0.05 was considered statistically significant. The analyses were performed using STATA software, version 17 (StataCorp. 2021 Statistical Software: Release 17; College Station, TX, USA, Stata Corporation).

### 2.5. Ethical Considerations

Subjects participating in the study were asked for informed consent. Confidentiality was ensured throughout all phases of the study. Personal data were anonymized and stored in REDCap software (Nashville, TN, USA), whose access was only granted to the members of the study in charge of data analysis. Data were aggregated before the analysis. All data were collected and treated according to the EU General Data Protection Regulation (GDPR) 2016/679.

## 3. Results

### 3.1. Demographics and Socioeconomic Characteristics

A total of 17,781 students and personnel, including 16,904 undergraduate and graduate students, 117 research fellows, 318 technical/administrative staff, and 382 researchers and professors were invited to participate in the survey. Overall, 1658 (9.3%) of the subjects entered the survey, whereas 1521 (8.5%) provided informed consent and completed it. Most of the participants were female (68.6%) and of Italian nationality (94.2%), in accordance with the sex and nationality distribution of the UPO population. Overall, most participants were represented by undergraduate and graduate students (77.9%), followed by researchers and professors (15.3%), and technical/administrative staff (6.8%). A significantly higher response rates were observed among professors (54.6%; *p* < 0.0001), researchers and research fellows (30.4%; *p* < 0.0001), technical/administrative staff (32.4%; *p* < 0.0001), and PhD students (12.2%; *p* = 0.001) compared to undergraduates (6.9%) [Table 1a].

Significantly higher than expected proportions of respondents were observed among medical biotechnology (*p* < 0.0001), nursing and obstetrics (*p* = 0.008), biotechnology (*p* < 0.0001), and medicine and surgery (*p* = 0.0008) students. On the contrary, business administration students demonstrated a significantly lower proportion of respondents than expected (*p* < 0.0001) (Table 1b).

### 3.2. Interest in Science and Scientific Research

Overall, participants showed a high interest in scientific disciplines and a deep trust in biomedical research and researchers (median = 9, IQR 8–10). A statistically significant difference in trust in biomedical research was found between the pre-COVID-19 period and during the COVID-19 pandemic (difference between pre- and post-COVID-19: *p* < 0.0001). Respondents rated the impact of scientific research on society as very important (median = 10, IQR = 9–10). Responses to the “risks and benefits of biomedical research” are detailed in Table 2. Most participants in the study (85.1%) stated that the benefits of biomedical research far outweigh the risks.

### 3.3. Biobanks and Their Activities

Knowledge about biobanks and biobanking activities is summarized in Table 3. This section was briefly introduced by an explanation about biobanks and their role in research. More than half of the participants (65%) declared to have heard the term “biobank” before this survey and knew what a biobank does. The knowledge level about biobanks and research biobanks was significantly higher among university personnel compared with students (*p* < 0.0001). Most of the respondents (84%) knew about the presence of research biobanks in Italy.

Participants demonstrated a high level of agreement with the sentence “research biobanks activities are important for the progress of biomedical research” (median = 9, IQR 8–10). In order to assess the level of trust in biobank activity, subjects were asked about their willingness to participate in biobanking. The majority of respondents (76.3%) were willing to provide biological samples to biobanks for research activity. The main reasons behind this choice were contributing to scientific research (75.3%), increasing knowledge in science for future generations (56.9%), and a sense of duty (23.2%). Only 5.3% of respondents, predominantly students, said they were unwilling to provide a biological sample to a biobank. None of them had any previous knowledge of biobanks.

To understand the factors that may affect participation in biobanks and biomedical research, respondents were questioned about concerns regarding the biobanking of biologic materials. Greater concerns were raised about the confidentiality of the information (25.6%), the commercial use of the samples (25.0%), and the fear of needles/blood (10.9%). Notably, almost all participants (96.0%) did not express any concern about genetic research.

Then, the willingness to provide personal information was explored. The majority (67.3%) of the participants were willing to provide health and lifestyle information, whereas only 5.5% of them were not inclined to give any information other than the biological samples.

Therefore, participants were asked about the level of information and control they would like to have over the use of their samples and data. Around half of the participants (43.2%) would like to know what research their sample would be used in, whereas 35.6% of respondents would prefer to be asked whenever their sample is used in a study.

## 4. Discussion

Our results clearly showed a high response rate among researchers and technical/administrative personnel and a very low response rate among students. The low rate among students could be explained by: the use of e-mail to administer the survey, the low interest in taking part in surveys, and the period of the study between August and December, a transitional period between the end of one academic year and the next. Higher rates were shown among scientific courses (biotechnology, biology, medicine, and surgery), as shown in other studies [5].

Overall, the majority (65%) of respondents among UPO students and personnel had a noteworthy level of knowledge of the meaning of the term “biobank” and the role that biobanks play in biomedical research. These results are in line with a study conducted in Jordan [5] in which 53% of university students had previous knowledge of the term biobank and of biobanking activities, whereas other studies conducted in Saudi Arabia and Russia reported levels of knowledge lower than 30% [14,15]. The high level of knowledge among the UPO population may be the consequence of local social media campaigns (including articles, interviews, and news) and, possibly, of their involvement in biobank research projects.

Most participants recognized the role of biobanks in research and, for this reason, were eager to provide biological samples to a biobank. In contrast with other studies [16], we did not find an association between sex and willingness to donate.

According to the results of a recent literature review [3], the advancement of biomedical research and benefits for society and future generations were the major reasons for the positive attitude toward participating in a biobank.

Concern about privacy, the risk of discrimination, the commercialization of samples, and associated data for profit seem to be major contributors to the hesitation in participating in a research biobank [3]. A quarter of participants raised a concern about the confidentiality of information (privacy) and the commercialization of biological samples. These were also the main concerns expressed in other studies [6,17,18,19,20], especially when genetic data are involved [13]. In our study the percentage of subjects willing to participate in biobanking decreased by about 10% when respondents were asked if they were willing to provide, in addition to biological samples, health and lifestyle data.

A biobank should inform healthcare professionals and the general population about the regulation under which they are treated and protected. Biobanks all over Europe are required to comply with the GDPR [21], which is widely implemented to address these concerns and grants protection to all EU citizens whenever their personal data are processed, with specific attention and derogations to health and genetic data processing (article 89 of the GDPR).

Of a different nature is the concern regarding the exploitation of samples for commercial purposes because participants consider such a purpose conflicting with their entirely disinterested choice (23.3%). In general, a higher trust in academic institutions than in private and/or profit organizations and biobanks has been reported [22,23,24,25], corroborating the key role of public academic institutions in contributing to scientific progress and health promotion but with the priority of protecting the human and legal rights of individuals. The UPO Biobank is a “non-profit service structure”, but this does not exclude that the obtained results can be exploited by private companies to improve health technologies and prepare new drugs because the population can benefit from the research only through these steps.

For all these reasons, a population biobank must match the ‘trust’ of citizens with the ‘trustworthiness’ of its governance: trustworthiness is an intrinsic ethical value and is also instrumental in increasing research participation and improving the perception of research by the public.

Genetic research entails several concerns and potential risks for individuals; surprisingly, a very small percentage of respondents (4.0%) had concerns about genetic research and the only one was “the fear of discovering a genetic disease/being a carrier of a genetic disease” (2.3%). Genetic research, such as genomic and genetic profiling, could result in the discovery of information on rare or complex conditions beyond the research target, also called “incidental findings” (IFs) [26,27]. The chance of identifying significant or pathogenic IFs using whole genome sequencing (WES) has been reported to be around 1–6% in the adult population [28,29], and this percentage is likely to increase. Deciding how and when to report IFs has been thoroughly discussed among scientists. National and international laws and guidelines provide minimal clues to researchers [26]. For example, the American College of Medical Genetics and Genomics has identified a list of genes that should be reported as IFs [30,31], whereas other authors in the UK suggest that IFs should be returned only in the case of serious conditions [32,33]. Surveys involving patients and the general population report mixed results: in the USA, IFs have been well-received by patients, while opposite results were found in a similar European study [34].

Participants show a high interest in scientific research (75.3%) and in providing a benefit for future generations (56.9%). At the same time, many respondents would prefer to be informed for every use of the donated sample (35.5%, asking for a specific consent). This may indicate the need to maintain some control over the type of research their sample and information are used in.

Therefore, the custody of biological samples needs to be nurtured through a continuous ‘connection’ with the biobank, understood as a physical place for storing and processing samples and above all as a ‘research community’ that binds researchers and citizens in a bond of solidarity and mutual responsibility. The new means of information make it possible to ensure that each sample trustee has a personalized flow of information on possible collaborations and different projects within the previously selected research areas, allowing the participant to exercise their rights at any time or to oppose processing operations that do not comply with their reference values. These are the characteristic features of the “dynamic” consensus which, currently is the most adequate and flexible response to the needs of “connection” manifested by citizens.

The main limitation of this study is the limited generalization of results to the general population. As with many other similar studies, it involved a selected academic population, and the response rate, even if higher than those in many other studies, was quite low. The few studies involving the general population seem to show results that are not far from those coming from selected populations [17,20,35]. This suggests that the estimations made by our study could be used, even if with some degree of caution, to plan the involvement of the population in cohort studies based on biobanking.

In the last 10 years, around 60 biobank-related surveys have been conducted and published all over the world, with less than 20 being related to European biobanks. In Italy, few studies have been conducted in the last 10 years [16,36,37], with none being published in the last 3 years. Of the three studies conducted in Italy, just one [16], published in 2017, investigated the willingness to donate and the attitude towards biobanking among university students. In comparison, we found a higher willingness to donate (76.3% compared to 57.7%), and a higher percentage knew about the presence of biobanks in Italy (84% compared to 43%). For the first time in years, this survey gives insight into the knowledge and perception of Italian university personnel about biobanks and scientific research and highlights critical issues regarding the perception that citizens’ have about the protection of their personal data and the type of informed consent to be offered to participants.

## 5. Conclusions

In this article, we reported high levels of support and willingness to donate and contribute to a research biobank of students and personnel of the University of Piemonte Orientale. Participants were aware of the fundamental role that biobanks play in research and were keen to participate in them for the sake of advancing biomedical research. Nonetheless, a number of concerns need to be addressed regarding the confidentiality of the data and information along with the commercial use of the samples and associated data. A well-structured survey aimed at the general population and investigating particular aspects of biobanking, including the return of results, privacy, data sharing, informed consent, and the commercial use of samples and data, could better clarify how to address these concerns.

## Figures and Tables

**Table 1 ijerph-19-13041-t001:** (**a**) Demographic characteristics of UPO students and personnel and those of participants. (**b**) Distribution of degree programs among UPO students.

(**a**)			
**Respondents**		**UPO Students and Personnel**	***p*-Value**
**Characteristic**	***n* (%)**	* **n** *	
Sex			
Male	477 (6.7)	7158	--
Female	1044 (9.8)	10,623	<0.0001
University role			
Students	1155 (6.9)	16,658	-
Ph.D. students	30 (12.2)	246	0.001
Research fellows	42 (23.7)	177	<0.0001
Researchers	49 (40.2)	122	
Professors	142 (54.6)	260	<0.0001
Administrative/technical staff	103 (32.4)	318	<0.0001
**Total**	1521 (8.5)	17,781	
(**b**)			
	**Respondents**	**UPO Students**	***p*-Value**
	***n* (%)**	***n* (%)**	
Business administration	81 (7.0)	2312 (13.9)	<0.0001
Biological science	157 (13.6)	2131 (12.8)	0.62
Biotechnology	220 (19.0)	1411 (8.5)	<0.0001
Medicine and Surgery	114 (9.9)	1007 (6.0)	0.0008
Medical Biotechnology	53 (4.6)	188 (1.1)	<0.0001
Nursing and Obstetrics	25 (2.2)	135 (0.8)	0.008
Philosophy	6 (0.5)	42 (0.2)	0.22
**Total**	1155	16,658	

**Table 2 ijerph-19-13041-t002:** Perception of benefits and risks associated with biomedical research.

Balance between Benefits and Risks	UPO Students	Personnel	Total
	*n* (%)	*n* (%)	*n* (%)
The benefits are greater than the risks.	908 (83.5)	290 (90.6)	1198 (85.1)
The benefits are slightly greater than the risks.	94 (8.6)	16 (5.0)	110 (7.8)
The benefits are equal to the risks.	72 (6.6)	11 (3.4)	83 (5.9)
The risks are slightly greater than the benefits.	5 (0.5)	1 (0.3)	6 (0.4)
The risks are greater than the benefits.	9 (0.8)	2 (0.6)	11 (0.8)
**Total**	1088 (100)	320 (100)	1408 (100)

**Table 3 ijerph-19-13041-t003:** Knowledge about biobanks and biobanking activities, concerns, and informed consent.

Questions	Students *n* (%)	Personnel *n* (%)	Total *n* (%)	*p*-Value
**Have you ever heard of the term research biobanks before?**				
Never	349 (29.5)	54 (16.1)	403 (26.5)	<0.0001
Occasionally	574 (48.4)	172 (51.2)	746 (49.0)	
Often	152 (12.8)	92 (27.4)	244 (16.0)	
Missing	110 (9.3)	18 (5.3)	128 (8.4)	
**Do you know which is the activity of research biobanks?**				
No	351 (29.6)	56 (16.7)	407 (26.8)	<0.0001
Yes, vaguely	404 (34.1)	106 (31.5)	510 (33.5)	
Yes, quite	288 (24.3)	111 (33.0)	399 (26.2)	
Yes, very well	31 (2.6)	45 (13.4)	76 (5.0)	
Missing	111 (9.4)	18 (5.4)	129 (8.5)	
**Do biobanks contribute to the advancement of biomedical knowledge?**				
Medium score (Range 0–10) [IQR]	9 [8,9,10]	9 [8,9,10]	9 [8,9,10]	0.0006
**Are you willing to donate biological samples to a biobank?**				
Yes	896 (75.6)	265 (78.9)	1161 (76.3)	0.15
Yes, under some conditions	95 (8.0)	36 (10.7)	131 (8.6)	
No	68 (5.7)	13 (3.9)	81 (5.3)	
Missing	126 (10.6)	22 (6.5)	148 (9.7)	
**Why would you donate a biological sample to a biobank?** (more than one option)				
Contribute to scientific research	888 (74.9)	257 (76.5)	1145 (75.3)	--
Increase knowledge/benefit for future generations	687 (58.0)	178 (53.0)	865 (56.9)	
Sense of duty	282 (23.8)	71 (21.1)	353 (23.2)	
Benefit for a family member or friends	210 (17.7)	24 (7.1)	234 (15.4)	
Personal benefit	186 (15.7)	22 (6.5)	208 (13.7)	
**What are your concerns about donating biological samples to a research biobank?** (more than one option)				
Fear that the confidentiality of the information is not guaranteed	318 (26.8)	71 (21.1)	389 (25.6)	--
Fear that my biological sample is being used for commercial purposes	305 (25.7)	75 (22.3)	380 (25.0)	
Fear of needles/blood	147 (12.4)	19 (5.6)	166 (10.9)	
Lack of confidence in the usefulness of my sample for scientific research	66 (5.6)	10 (3.0)	76 (5.0)	
Fear of genetic research	49 (4.1)	20 (6.0)	69 (4.5)	
Little confidence in the quality of scientific research using my sample	53 (4.5)	14 (4.2)	67 (4.4)	
No time	47 (4.0)	17 (5.1)	64 (4.2)	
No concerns	377 (31.8)	134 (40.0)	511 (33.6)	
**What form of consent would you give for the use of your biological sample if you choose to commit it to a research biobank?**				
I would like to be asked for consent whenever a study will need to use my sample.	448 (37.8)	93 (27.7)	541 (35.6)	<0.0001
I would consent to the use of my sample for research, about which I would like to be informed, but without need to provide further consents.	494 (41.7)	163 (48.5)	657 (43.2)	
Once I have given the sample, I am not at all interested in knowing if and how it will be used.	47 (4.0)	36 (10.7)	83 (5.5)	
I do not know.	26 (2.2)	17 (5.1)	43 (2.8)	
Missing	170 (14.3)	27 (8.0)	197 (12.9)	
**If you decide to donate a biological sample to a research biobank, would you also provide personal information?**				
Yes, I would also provide information regarding health, lifestyle, and habits	787 (66.4)	237 (70.5)	1024 (67.3)	0.20
Yes, but I would only provide information about health and not information related to habits and lifestyle.	107 (9.0)	25 (7.4)	132 (8.7)	
No, I would only provide the biological sample.	64 (5.4)	19 (5.7)	83 (5.5)	
I do not know.	64 (5.4)	29 (8.6)	93 (6.1)	
Missing	163 (13.8)	26 (7.7)	189 (12.4)	

## Data Availability

Not applicable.

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
