# Peer review of "Public Attitude towards Biobanking: An Italian University Survey"

_ijerph, 2022, doi:10.3390/ijerph192013041_

Round 1
Reviewer 1 Report
In this study, based on online questionnaire, Chiara et. al performed an investigation on knowledge level and perception of biobanks in students and personnel of the University of Piemonte Orientale. They found the majority of participants knew the importance of biobanks, while concerns from commercial usage cannot be ignored. Public attitude is a very important factor for project design and approval. Here, I’m happy to see that a lot of participants were willing to provide biospecimens and/or health data. In general, I think this is an interesting study which may provide potential guidance to project funders.
The following issues should be considered and addressed clearly:
In the “Statistical analysis” of Method Section, the authors mentioned that “Analyses were performed using STATA software”. But they didn’t mention what type of the statistic have been done. Similarly, in line 142, 144, 147, 148, readers won’t know what test they have done and what method the Pvalue come from. In line 158 to 160, readers cannot even find a Pvalue. All Pvalues mentioned in the main text should be also presented clearly in the corresponding table.
Reviewer 2 Report
Aleni and colleagues present results of an electronically-administered survey assessing knowledge and perception toward biobanking among personnel, students, faculty, etc. of an Italian academic institution. As Biobanks grow in scientific utility, participation, and public interest, this report is a timely and important addition to the literature. However, some changes and enhancements are needed to the methods and results to improve the quality and value of this manuscript for scientific consumption.
Major comments
1. Enhancement of statistical approach and reporting.
a. Need to have better consistency between methods and results. The statistical analysis section of the methods should describe the statistical comparisons made—pvalues are reported in reference to Table 1, Table 2, and Table 3 comparisons but no methods are described in the methods section and it is unclear which groups are being compared. For example, in line 142 “a significant high response rate was observed among professors (54.6%; p<0.0001)”—high among professors compared to whom? Include comparisons being made in methods (and with what statistical test) and make more clear what the comparison group is when reporting results. 54.6% does not appear in Table 1—if you want to discuss row proportions in the text, consider reporting those in the table too, instead of column proportions. Similar confusion around what the statistical comparison is for lines 146-147 “…was significantly different than expected, according to the profile…(p<0.001).” What was the expectation compared against? Student response rate vs. faculty response rate? Significantly different from 0?
b. Tables and results text should include Ns, %, and p-values (where statistical comparisons were made) for all results reported. In most surveys, participants may not answer every question and so the N and % are important to report as they may change from question to question. Table 1.b. needs a second column (n, %) with students invited to respond by degree program. Table 2 needs Ns in addition to %.
c. To be most useful to the biobanking community, the differences in Table 2 and 3 results by student/personnel group (student, researchers, professors, etc.) would be useful—for example, who (which group of people) needs more information or extra attention to improve trust in biobanking? Lines 170-173 suggest the analysis has been run, since it has a pvalue provided; these analyses/results should be described in the methods, and reported in the results by adding columns for each group to Table 2 and Table 3 results.
2. How have results presented here changed over time? As pointed out by the authors in lines 305-306, it has been several years since a similar survey has been published from an Italian population. There should be at least some comparison (discussion) of these newer findings with those presented in the last Italian report in this space. Changes to knowledge and perceptions of biobanking in a similar target population over time would be useful as participation and public knowledge of biobanking continues to increase.
Minor comments:
1. Include the overall response rate in the abstract, including: denominator of invited participants, numerator of respondents, and %
2. Lines 111-117 have some formatting issues.
3. Great job translating some relevant questions to English for inclusion in Table 3. Rather than having to translate the full questionnaire, provide more information in the methods on the number of total items and the number of items per section (lines 92-97).
4. The reporting of pre- vs. post-COVID19 results is confusing because it has not yet been introduced. Perhaps include in the methods more information on asking about covid19 before reporting results on it.
5. Please clarify the language and message in a couple of places—it’s unclear what the point of these sentences are: lines 69-73, lines 286-295.
6. Lines 198-209—does this belong in the introduction? Setting up the Aims seems more appropriate for intro than discussion.
7. Lines 228-230—where were these results presented in the results or methods? Please don’t discuss results that were not presented earlier in the paper.
Round 2
Reviewer 1 Report
I think the authors have answered my questions.I have no more questions.